# Attribute-Driven Multimodal Hierarchical Prompts for Image Aesthetic Quality Assessment

## ABSTRACT

Image Aesthetic Quality Assessment (IAQA) aims to simulate users' visual perception to judge the aesthetic quality of images. In social media, users' aesthetic experiences are often reflected in their textual comments regarding the aesthetic attributes of images. To fully explore the attribute information perceived by users for evaluating image aesthetic quality, this paper proposes an image aesthetic quality assessment method based on attribute-driven multimodal hierarchical prompts. Unlike existing IAQA methods that utilize multimodal pre-training or straightforward prompts for model learning, the proposed method leverages attribute comments and quality-level text templates to hierarchically learn the aesthetic attributes and quality of images. Specifically, we first leverage users' aesthetic attribute comments to perform prompt learning on images. The learned attribute-driven multimodal features can comprehensively capture the semantic information of image aesthetic attributes perceived by users. Then, we construct text templates for different aesthetic quality levels to further facilitate prompt learning through semantic information related to the aesthetic quality of images. The proposed method can explicitly simulate users' aesthetic judgment of images to obtain more precise aesthetic quality. Experimental results demonstrate that the proposed IAQA method based on hierarchical prompts outperforms existing methods significantly on multiple IAQA databases. Our source code is provided in the supplementary material, and we will release all source code along with this paper.

## CCS CONCEPTS

• **Computing methodologies** → **Image representations**.

## KEYWORDS

Image aesthetic quality assessment, aesthetics-driven, multimodal learning, hierarchical prompts

## 1 INTRODUCTION

Nowadays, with the widespread popularity of social media platforms, images have become one of the mainstream media of communication and expression in the digital age. From Instagram to WeChat, billions of photos are shared by users globally every day, reflecting their experiences, emotions, and artistic inclinations. The

*ACM MM, 2024, Melbourne, Australia*
© 2024 Copyright held by the owner/author(s). Publication rights licensed to ACM.
ACM ISBN 978-x-xxxx-xxxx-x/YY/MM
https://doi.org/10.1145/nnnnnnn.nnnnnnn

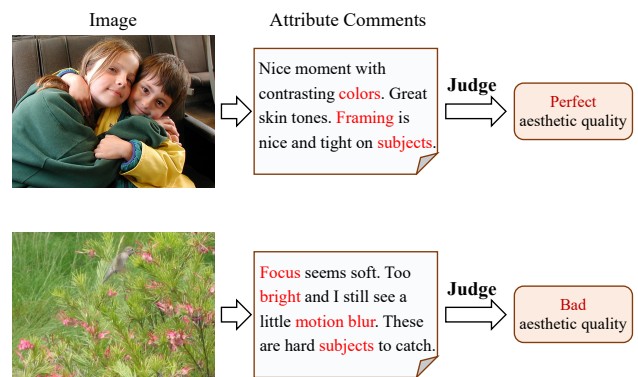

Image      Attribute Comments

Nice moment with contrasting colors. Great skin tones. Framing is nice and tight on subjects.

Judge → Perfect aesthetic quality

Focus seems soft. Too bright and I still see a little motion blur. These are hard subjects to catch.

Judge → Bad aesthetic quality

**Figure 1: Two images collected from social media for the AVA database [30], along with corresponding aesthetic comments and quality from users [48]. Users first evaluate the aesthetic attributes of images through textual comments. Then, users can explicitly judge the aesthetic quality of images based on these attribute comments. (a) An image with perfect aesthetic quality. (b) An image with bad aesthetic quality.**

abundance of visual content highlights the importance of understanding and evaluating image aesthetic quality in an automated manner. In light of this, researchers in the fields of image processing and multimedia content experience have shown great interest in exploring image aesthetic quality assessment (IAQA) that can simulate users' aesthetic perception to evaluate the quality of images [4]. IAQA methods have significant application value in various areas, including image recommendation [38], photo retrieval [23], image enhancement [5], photo cropping [24], as well as image stylization [42].

In earlier years, IAQA models mainly relied on hand-crafted features designed based on photography rules and aesthetic attributes, such as composition, color, lighting, etc [3]. Recently, deep learning methods have been widely applied in IAQA tasks because of their powerful feature representation ability [25]. In general, existing IAQA methods typically leverage aesthetic attributes to assist in learning image aesthetic quality, which demonstrates superior performance [17, 20]. However, these methods typically utilize numerical attributes or related implicit features to obtain assessment models for image aesthetic quality. In addition, they directly adopt deep models to learn the mapping relationship between images and aesthetic attributes or quality [11, 49], which makes it challenging to effectively extract the relevant information about users' aesthetic perception of images. Therefore, we need to introduce more modal data (such as aesthetic-related textual information) to fully represent users' aesthetic judgment of images.

In recent years, multimodal learning methods have been widely applied in various visual domains due to their ability to learn rich semantic information [16, 46, 48]. Large pre-trained vision-language models, such as CLIP [34], have gained increasing attention. These models, trained on massive image-text pairs, are highly efficient in downstream unimodal or multimodal tasks [9]. Therefore, multimodal prompt learning-based IAQA models have also been proposed in the past two years. One approach is to directly obtain a pre-trained model from multimodal learning and fine-tune it in the task of image aesthetic quality assessment [36]. Another approach is to build an IAQA model by using simple quality prompts [41]. Although multimodal learning methods have shown promising results in IAQA tasks, these methods have not fundamentally revealed the critical factors that affect users' aesthetic judgments of images. Generally, users' judgments on the aesthetic quality of images are a gradual process that can be divided into two stages. For instance, Fig. 1 demonstrates two images collected from social media for the AVA database [30], along with corresponding aesthetic comments and quality from users [48]. As can be seen from this figure, users first evaluate the aesthetic attributes of images through textual comments and then can leverage this attribute information to explicitly judge the aesthetic quality of images. Consequently, to accurately measure the aesthetic quality of images, it is necessary to design the above two-step prompts to model the process of users' aesthetic judgments of images.

Therefore, this paper proposes an attribute-driven multimodal hierarchical prompts learning method for image aesthetic quality assessment, which is abbreviated as AMHP. First of all, users usually comment on the aesthetic attributes of images on social media, which makes it possible to obtain text descriptions related to attributes from users' comments [48]. If user comments are unavailable, we can also construct corresponding attribute text templates through labeled attributes [17, 43]. To fully reveal the semantic information of image aesthetic attributes, we utilize attribute comments to capture the attribute-driven multimodal features of aesthetic aspects in images through prompt learning. Then, we construct text templates as the prompts for different levels of image aesthetic quality. Concretely, we leverage cosine similarity to calculate the relevance between text template features of different quality levels and attribute-driven multimodal features to obtain the weights for each aesthetic quality level. Finally, guided by the hierarchical prompts of attribute comments and text templates of quality levels, we derive the aesthetic quality score by assigning weights to each quality level. The proposed method can more precisely predict the aesthetic quality of images perceived by users through the efficacious and comprehensible attribute-driven multimodal hierarchical prompts strategy.

To sum up, the contributions of the proposed IAQA method are three-fold.

- We propose an attribute-driven multimodal learning strategy. By embedding aesthetic attribute comments with images, we can utilize more comprehensive semantic information of aesthetic attributes for modeling the aesthetic quality of images. Moreover, our method can also learn an effective IAQA model by constructing attribute text templates in the absence of user comments on images.

- We propose a multimodal hierarchical prompts learning approach. By hierarchically utilizing text descriptions of aesthetic attributes and different quality levels for multimodal prompted learning, the proposed method can effectively simulate users' aesthetic judgments of images in actual situations.

- We propose an image aesthetic quality assessment method based on attribute-driven multimodal hierarchical prompts learning. Extensive experiments and comparisons are conducted on three mainstream IAQA databases, and experimental results demonstrate that the proposed method outperforms state-of-the-art IAQA methods.

## 2 RELATED WORK

In this section, we first review some works on IAQA (Section 2.1) and then introduce the related works of multimodal prompt learning (Section 2.2).

## 2.1 Image Aesthetic Quality Assessment

Existing IAQA methods for evaluating image aesthetic quality can be divided into three major tasks [45, 47] according to different objectives, i.e., aesthetic binary classification [12, 31], aesthetic score regression [17, 21] and aesthetic distribution prediction [8, 10]. Early methods [3, 18, 28, 40] mainly leveraged hand-crafted features to represent the photographic rules present in computational images, the global layout of images, and typical objects in images. Tang *et al.* [40] proposed to extract visual features based on image content for image aesthetic quality assessment. Kucer *et al.* [18] demonstrated that combining a series of hand-crafted features can achieve significant improvement in predicting image aesthetic quality. Marchesotti *et al.* [28] used the descriptors to aggregate statistics computed from hand-crafted features for assessing the aesthetic quality of photographs. Although these IAQA methods based on hand-crafted features attempt to predict image aesthetic quality through photographic rules and have achieved certain success, these hand-crafted features can not comprehensively reveal the aesthetic characteristics of images due to their limited representation ability.

With the emergence of large-scale image aesthetic quality assessment databases [30], traditional methods based on hand-crafted features face challenges in handling large amounts of training data. Due to its powerful ability in feature representation, deep learning has been adopted by recent IAQA methods [1, 35]. Existing deep learning-based methods typically utilize attribute-related deep features to assist in building an image aesthetic quality assessment model. For instance, Kong *et al.* [17] proposed an AADB database, which included 11 aesthetic attributes, and leveraged the deep features based on attributes and content to rank the aesthetic quality of images. Celona *et al.* [1] utilized deep features related to aesthetic attributes such as style and composition to automatically adapt the hyperparameters of the proposed image aesthetic distribution prediction network. She *et al.* [35] presented a unified IAQA method based on layout attributes, which extracts implicit attribute features through a layout-aware graph convolutional module to evaluate the aesthetic quality of images. In addition, Yang *et al.* [43] constructed an IAQA database that incorporates a rich set of aesthetic attributes, and demonstrated that these aesthetic attributes can

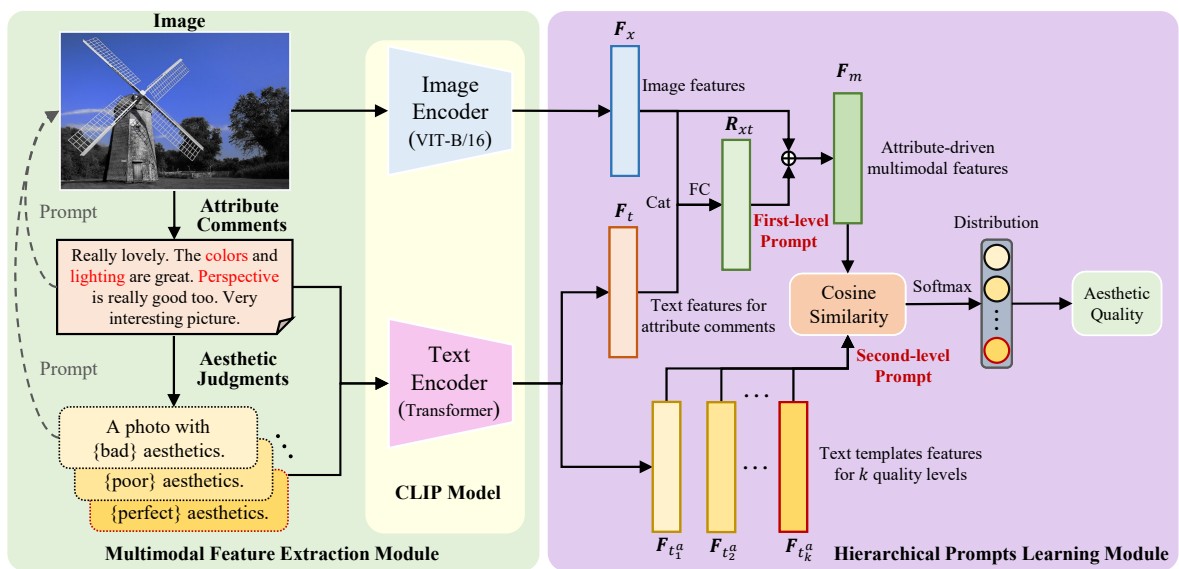

**Figure 2: The technical pipeline of our model. The proposed model includes two parts: a multimodal feature extraction module and a hierarchical prompts learning module. In the first part, we utilize the image and text encoders from the CLIP model to extract multimodal features of images and corresponding attribute comments, as well as aesthetic judgments of different quality levels. In the second part, we introduce hierarchical prompts learning to simulate users' aesthetic quality judgment of images.**

effectively enhance the performance of IAQA models and improve their interpretability. Although these IAQA methods have shown promising performance in extracting aesthetic attribute features for modeling image aesthetic quality, these features are driven by numerical attributes [17, 33, 50] and fail to effectively reveal the semantic information of aesthetic attributes [32], leading to insufficient representation of the aesthetic quality of images. Therefore, it is necessary to introduce semantic features that can represent aesthetic attributes to evaluate the aesthetic quality of images.

## 2.2 Multimodal Prompt Learning

In recent years, multimodal learning has made significant progress in the fields of computer vision and image processing by utilizing cross-modal correlation features to expose more semantic information [44]. In addition, multimodal prompt learning can learn effective visual models by introducing text prompt templates [34]. Radford *et al.* [34] proposed a visual-language pre-trained CLIP model, which enables multimodal features to better represent semantic information through contrastive learning of 400 million image-text pairs. However, performing prompts only in one modality (image or text) may only achieve sub-optimal performance. Hence, Khattak *et al.* [15] proposed embedding prompts in each modality branch in multimodal learning to enhance semantic consistency between visual and language representations. This approach demonstrates that richer and more comprehensive semantic prompts can lead to better performance.

Recently, multimodal prompt learning has also been applied in some IAQA methods due to its semantic representation ability [14, 36, 41]. For example, Wang *et al.* [41] applied the pre-trained

CLIP model to the task of image aesthetic quality assessment by designing a simple paired prompt template. Ke *et al.* [14] learned image aesthetics from user comments on social media and proposed a multimodal aesthetic representation method based on visual language pre-training. Besides, Sheng *et al.* [36] proposed a pre-trained model for image aesthetic quality evaluation based on contrastive learning from multiple attributes of user comments. Although existing multimodal learning-based methods can obtain more aesthetic-related semantic information, these methods still directly leverage straightforward prompts or pre-trained models to predict the aesthetic quality of images [14, 36, 41], and cannot explicitly simulate users' aesthetic perception in judging the quality of images. Therefore, it is necessary to utilize multimodal prompt learning to accurately model the process of users evaluating the aesthetic quality of images.

## 3 PROPOSED METHOD

### 3.1 Overview

In this section, we introduce the proposed IAQA method based on attribute-driven multimodal hierarchical prompts learning (AMHP). As shown in Fig. 1, the process of users' aesthetic judgment on images can be divided into two stages. In light of this, we design two prompts for multimodal learning. Firstly, we extract text descriptions related to aesthetic attributes from user comments on images as the first-level prompt. Then, we design text templates of different aesthetic quality levels as the second-level prompt. Consequently, the proposed attribute-driven multimodal hierarchical prompts can explicitly simulate the process of users judging image aesthetic quality.

In Fig. 2, we show the technical pipeline of the proposed AMHP model which is composed of two parts, namely, a multimodal feature extraction module and a hierarchical prompts learning module. In the multimodal feature extraction module, we first obtain attribute comments based on keywords related to image aesthetic attributes (such as color, lighting, composition, and theme), and then design a corresponding set of text templates for aesthetic judgments based on different levels of image aesthetic quality. Finally, we utilize the image encoder and text encoder from the CLIP model [34] to extract the aforementioned multimodal features. In the hierarchical prompts learning module, we first leverage the textual features of attribute comments to perform prompt learning on image features, and then fuse them to obtain attribute-driven multimodal features. Then, the text template features of aesthetic judgments at different quality levels are further used for prompt learning on the attribute-driven multimodal features, aiming to accurately simulate users' aesthetic judgments of images. In the subsequent sections, we present these two parts in detail.

### 3.2 Multimodal Feature Extraction Module

To conduct the proposed multimodal hierarchical prompts learning, we need to jointly extract the multimodal features of images and attribute comments. In addition, text template features for aesthetic judgments of different quality levels should be extracted. Therefore, in this module, we adopt the pre-trained CLIP model [34] as the backbone network, which can leverage an image encoder based on VIT-B/16 and a text encoder based on Transformer (as shown in the left half of Fig. 2) for feature extraction. Specifically, for an image $x$ and a segment of text $t$, the image encoder $E_{\theta_x}$ and the text encoder $E_{\theta_t}$ can map them to two features with the same dimension

$$F_x = E_{\theta_x}(x), F_t = E_{\theta_t}(t), \qquad (1)$$

where $F_x \in \mathbb{R}^n$ and $F_t \in \mathbb{R}^n$ denote image features and text features, respectively. $\theta_x$ and $\theta_t$ represent the parameters of the image encoder $E_{\theta_x}$ and the text encoder $E_{\theta_t}$.

In particular, we suppose that $\{x_i\}_{i=1}^{N_{tr}}$ represents the training images collected in an IAQA database, where $N_{tr}$ denotes the total number of training images. These images in the database are usually collected from social media (such as the DPChallenge website), and users can comment on the aesthetic attributes of these images (as shown in Fig. 2). Therefore, the aesthetic attribute comments of users on these images can also be obtained. We assume that $\{t_i\}_{i=1}^{N_{tr}}$ represents the corresponding attribute comments of these images. Based on the image encoder and text encoder of the pre-trained CLIP model, we can obtain the multimodal features of the image $x_i$ and text $t_i$, which can be calculated by $F_{x_i} = E_{\theta_x}(x_i)$ and $F_{t_i} = E_{\theta_t}(t_i)$, where $F_{x_i}$ and $F_{t_i}$ indicate the features of the $i$-th image and text, respectively.

In our proposed multimodal prompt learning, in addition to extracting text features of attribute comments for prompting, we also need to obtain the aesthetic quality of images based on users' aesthetic judgment of images. To this end, we design a set of text prompt templates based on different aesthetic quality levels. We assume that $k$ represents the number of total quality levels. Then, we can obtain $k$ text prompt templates, which are "A photo with {quality − level} aesthetics.", where {quality − level} includes all

quality levels of users aesthetic judgments on images. For example, if $k = 5$, the {quality − level} can be "bad", "poor", "fair", "good", and "perfect". Therefore, the set of text templates with different quality levels can be represented as

$$T_a = \{t_1^a, t_2^a, t_3^a, \ldots, t_k^a\}. \qquad (2)$$

Through the text encoder $E_{\theta_t}$, we can obtain the features of the text templates for aesthetic judgments, which take the form

$$F_{T_a} = E_{\theta_t}(T_a), \qquad (3)$$

where $F_{T_a} = \{F_{t_1^a}, F_{t_2^a}, F_{t_3^a}, \ldots, F_{t_k^a}\}$ represents a series of text template features of different aesthetic quality levels. In this module, we utilize the image encoder and text encoder from the CLIP model to obtain the multimodal features $\{F_{x_i}, F_{t_i}\}_{i=1}^{N_{tr}}$ of training images and corresponding attribute comments, respectively. In addition, we also obtain the features $F_{T_a}$ of $k$ quality-level text templates through the text encoder. In this way, we extract image features and two aspects of text features to provide a foundation for hierarchical prompts learning.

### 3.3 Hierarchical Prompts Learning Module

To explicitly simulate the aesthetic judgment process of users on the aesthetic quality of images, we perform hierarchical prompts learning through attribute comments and text templates of quality levels in this module. Firstly, we leverage the prompt information of attribute comments to capture richer semantic features about aesthetic attributes from images. Inspired by the fusion of multimodal features in [2], we combine the multimodal features of the $i$-th image and its corresponding attribute comments, which is defined as

$$F_{xt_i} = Concate(Norm(F_{x_i}), Norm(F_{t_i})), \qquad (4)$$

where $Concate$ denotes the concatenation operation and $Norm$ is the l2-normalization operation. To further integrate with image features, we employ a fully connected layer $FC_\theta$ to map the feature $F_{xt_i} \in \mathbb{R}^{2n}$ to a $n$-dimensional feature $R_{xt_i} = FC_\theta(F_{xt_i})$. Next, we add the normalized image feature $F_{x_i}$ and the combined feature $R_{xt_i}$ to obtain the attribute-driven multimodal feature $F_{m_i} \in \mathbb{R}^n$ that can represent the aesthetic attributes of the $i$-th image, which can be calculated by

$$F_{m_i} = R_{xt_i} + Norm(F_{x_i}). \qquad (5)$$

Through the above-mentioned prompt learning of images based on attribute comments, the obtained attribute-driven multimodal features $\{F_{m_i}\}_{i=1}^{N_{tr}}$ can effectively reveal users' perceptual characteristics of image aesthetic attributes, and accurately capture the semantic information of users' aesthetic judgments on attributes.

In general, users can explicitly measure the aesthetic quality of images based on their informing on image aesthetic attributes. Therefore, we then leverage semantic information from text templates with different quality levels for prompt learning to achieve the final aesthetic quality. To achieve this, cosine similarity is adopted to calculate the relevance between the attribute-driven multimodal features of the $i$-th images and the text template features of each quality level, which takes the form

$$logit(q_j|x_i) = \frac{F_{m_i} \cdot (F_{t_j^a})^{\mathrm{T}}}{||F_{m_i}|| \cdot ||F_{t_j^a}||}, j \in \{1, 2, 3, \ldots, k\}, \qquad (6)$$

 

where $\text{logit}(q_j|x_i)$ represents the correlation coefficient between the multimodal features of the image $x_i$ and the $j$-th aesthetic quality level. Then, we apply the softmax function to calculate the probability distribution of aesthetic quality levels with a temperature parameter $\tau$, which takes the form

$$\hat{p}(q_j|x_i) = \frac{\exp(\text{logit}(q_j|x_i)/\tau)}{\sum_{j=1}^{k}(\exp(\text{logit}(q_j|x_i)/\tau)}, \quad (7)$$

where $\hat{p}(q_j|x_i)$ denotes the predicted probability of the $i$-th image at the $j$-th aesthetic quality level. Therefore, we can obtain the aesthetic distribution of all training images.

In the IAQA task, Earth Mover's Distance (EMD) is a commonly used loss function for calculating aesthetic distributions [39]. Hence, we utilize the EMD loss function to calculate the difference between the predicted probability of quality levels and the probability of quality levels rated by different users, which is defined as

$$\mathcal{L} = \left( \frac{1}{N_{tr}} \sum_{i=1}^{N_{tr}} \left| CDF_{\{\hat{p}(q_j|x_i)\}_{j=1}^{k}} - CDF_{\{p(q_j|x_i)\}_{j=1}^{k}} \right| \right)^{\frac{1}{r}}, \quad (8)$$

where $N_{tr}$ is the total number of training images, and $CDF$ denotes the cumulative distribution function. $\{p(q_j|x_i)\}_{j=1}^{k}$ represents the probability of quality levels rated by different users and $r$ indicates the penalty factor between two distributions. By training the proposed model with the loss function $\mathcal{L}$, we can obtain the proposed AMHP model that predicts the aesthetic distribution of images. The aesthetic quality score can be derived by assigning probability weights to each quality level in the aesthetic distribution of images.

During the testing phase, we input a test image and corresponding attribute comment $\{x_s, t_s\}$ into the trained AMHP model, which can obtain the aesthetic distribution $\{\hat{p}(q_j|x_s)\}_{j=1}^{k}$ of the image (distribution prediction). Furthermore, we can also calculate aesthetic quality score by weighted summation of the probability of different quality levels (score regression), and further divide the score into high and low categories (binary classification).

## 4 EXPERIMENTS

### 4.1 Image Databases

To evaluate the performance of our AMHP model, we conduct extensive experiments on the AVA database [30], the PARA database [43], and the AADB database [17].

The AVA database [30] consists of more than 250,000 images collected from the DPChallenge website. Each image is rated by approximately 210 users with aesthetic quality scores ranging from 1 to 10 and some images also contain semantic and style category labels. Besides, Zhou *et al.* [48] propose the AVA-Comments database that collects user comments from the DPChallenge based on the website addresses of images in the AVA database. Among all user comments, we can leverage attribute keywords (such as color, lighting, composition, and theme) to extract these comments containing aesthetic attributes from user comments, which serve as prompt information for aesthetic attributes. For images that do not contain attribute comments, we select one of these comments as the prompt information. Besides, the corresponding quality prompts are designed according to different levels of aesthetic quality. In

our experiments, The training and testing sets adopt the standard partitioning method of the database [30].

In the PARA [43] database, 31,220 images with abundant attributes are annotated by 438 users. Each image is annotated with aesthetic quality scores which range from 1 to 5 and attributes by some users. Images are labeled with six attributes (i.e., color, light, depth of field, composition, content, and object emphasis), the scores of which range from 1 to 5. Although this database does not contain user comments, we can construct text templates of the corresponding attributes based on the above attribute labels, which are used as prompt information for aesthetic attributes. For the five levels of attribute scores, we can construct five corresponding attribute text templates. For example, for the attribute of "color", the text template can be "A photo with {**attribute** − **level**} color.". The {**attribute** − **level**} also can be "bad", "poor", "fair", "good", and "perfect". Besides, the corresponding quality prompts are designed according to different levels of aesthetic quality. In our experiments, The training and testing sets adopt the standard partitioning method of the database [43].

In the AADB [17] database, 10,000 images are collected from the Flickr website. Each image contains aesthetic scores and aesthetic attributes. The rating scale for aesthetic quality scores is from 1 to 5. There are a total of eleven aesthetic attributes (i.e. interesting content, object emphasis, good lighting, color harmony, vivid color, depth of field, motion blur, rule of thirds, balancing element, repetition, and symmetry) and the score range for each attribute is from -1 to 1. Similar to the PARA database, we utilize these attributes to construct text templates, which are used as prompt information for aesthetic attributes. Besides, the corresponding quality prompts are designed according to different levels of aesthetic quality. In this database, we employ 9000 images for training and 1,000 images for testing.

### 4.2 Experimental Settings

*4.2.1 Implementation Details.* In the proposed model, we adopt the pre-trained CLIP model as the backbone, which applies VIT-B/16 as the image encoder and the Transformer as the text encoder [34]. The number of quality-level text templates $k$ is set to 5. In the training phase, all the parameters of our model are optimized by an AdamW optimizer with a decoupled weight decay regularization of $10 − 3$. We set $r$ to 2. The batch size is set to 72 and the total epoch for model training is set to 5. The source code of our model is based on PyTorch. All experiments are conducted on a single NVIDIA GeForce RTX 4090 GPU.

*4.2.2 Evaluation Criterion.* We evaluate the performance of our method on three IAQA tasks: aesthetic binary classification, aesthetic score regression, and aesthetic distribution prediction [35, 36, 39, 45]. Similar to the above IAQA methods, we adopt six common evaluation criteria to measure the performance of our method and existing IAQA methods. Specifically, Accuracy (ACC) is used for aesthetic binary classification. In addition, Spearman Rank-order Correlation Coefficient (SRCC), Pearson Linear Correlation Coefficient (PLCC), and Mean Squared Error (MSE) are employed to evaluate the performance of aesthetic score regression. For aesthetic distribution prediction, we leverage EMD with r = 1 (EMD1) and r = 2 (EMD2) to evaluate the performance of IAQA methods.

**Table 1: Performance comparison of the proposed model with the state-of-the-art IAQA models on the AVA database. "-" denotes unreported results.**

| Methods | ACC ↑ | SRCC ↑ | PLCC ↑ | MSE ↓ | EMD1 ↓ | EMD2 ↓ |
|---|---|---|---|---|---|---|
| DMA-Net [26] | 75.4 | - | - | - | - | - |
| Kong *et al.* [17] | 77.3 | 0.558 | - | - | - | - |
| NIMA [39] | 78.2 | 0.633 | 0.647 | 0.330 | 0.049 | 0.071 |
| APM [29] | 80.3 | 0.709 | - | 0.279 | - | 0.061 |
| A-Lamp [27] | 82.5 | - | - | - | - | - |
| Zeng *et al.* [45] | 80.8 | 0.719 | 0.720 | 0.275 | - | 0.065 |
| Hosu *et al.* [7] | 81.7 | 0.756 | 0.757 | - | - | - |
| MUSIQ [13] | 81.5 | 0.726 | 0.738 | 0.242 | - | - |
| TANet [6] | 80.6 | 0.758 | 0.765 | - | 0.047 | - |
| Calona *et al.* [1] | 80.8 | 0.732 | 0.733 | - | 0.044 | - |
| SAGAN [37] | 83.7 | 0.774 | 0.788 | - | - | - |
| Niu *et al.* [32] | 81.9 | 0.734 | 0.740 | 0.242 | - | - |
| CLIP [34] | 81.6 | 0.744 | 0.753 | - | - | - |
| VILA-R [14] | - | 0.774 | 0.774 | - | - | - |
| AesCLIP [36] | 83.1 | 0.771 | 0.779 | 0.218 | 0.041 | 0.058 |
| **AMHP** | **84.5** | **0.804** | **0.818** | **0.186** | **0.036** | **0.044** |

## 4.3 Comparison with State-of-the-Art Methods

In this subsection, we compare the proposed method with state-of-the-art IAQA methods on all three IAQA databases, including the AVA [30], PARA [43], and AADB [17] databases.

*4.3.1 Performance on the AVA database.* The attribute comments for each image in this database [30] can be provided by the AVA-Comments database [48]. Therefore, we can leverage attribute comments to train and test our AMHP model. Table 1 lists the comparison results of our method and state-of-the-art methods, and the best results for each criterion are shown in bold. From the table, we can see that our method yields the best performance in all three IAQA tasks. Specifically, our method outperforms IAQA methods based on visual features from single-modal images (such as Kong *et al.* [17], NIMA [39], Calona *et al.* [1] and SAGAN [37]) by a large margin, indicating that the proposed method can capture richer semantic information from attribute comments, which can be used to effectively assist aesthetic quality evaluation. Furthermore, our AMHP model is also superior to pre-trained multimodal IAQA models (VILA-R [14] and AesCLIP [36]) and straightforward prompts-based IAQA model (CLIP [34]), demonstrating that our attribute-driven hierarchical prompts learning strategy can achieve more promising performance by simulating users' judgment process of image aesthetic quality. In summary, the proposed AMHP model gradually captures semantic information of aesthetic attributes and quality through multimodal hierarchical prompts learning, resulting in a highly efficient IAQA model.

*4.3.2 Performance on the PARA database.* The PARA database [43] is a recently proposed database that contains a variety of labeled aesthetic attributes. In particular, the training images and constructed attribute text templates are used to train the proposed model and the trained model is then used to predict the aesthetic quality of all test images. Table 2 lists the comparison results between our AMHP method and several representative IAQA methods on the PARA database and these IAQA methods mainly leverage this database for aesthetic binary classification and score regression. As shown in this table, the best result is bolded and we observe that our method can achieve better performance than these IAQA methods that

**Table 2: Performance comparison of the proposed model with the state-of-the-art IAQA models on the PARA database.**

| Methods | ACC ↑ | SRCC ↑ | PLCC ↑ |
|---|---|---|---|
| NIMA (Resnet-50) [39] | 87.5 | 0.882 | 0.922 |
| PA_IAA (Densenet-121) [21] | 87.5 | 0.877 | 0.919 |
| MUSIQ [13] | 88.1 | 0.882 | 0.918 |
| TANET [6] | 89.2 | 0.883 | 0.917 |
| Yang *et al.* (Swin-T) [43] | 88.6 | 0.902 | 0.936 |
| TAVAR [20] | 89.7 | 0.911 | 0.940 |
| AesCLIP [36] | 89.9 | 0.926 | 0.951 |
| **AMHP** | **92.3** | **0.955** | **0.966** |

**Table 3: Performance comparison of the proposed model with the state-of-the-art IAQA models on the AADB database.**

| Methods | SRCC ↑ |
|---|---|
| RegNet [17] | 0.678 |
| Pan *et al.* [33] | 0.704 |
| NIMA (Resnet-50) [39] | 0.708 |
| RGNet [22] | 0.710 |
| Zhu *et al.* (Resnet-152) [50] | 0.716 |
| Zeng *et al.* [45] | 0.726 |
| HIAA [19] | 0.739 |
| Calona *et al.* [1] | 0.757 |
| TAVAR [20] | 0.761 |
| SAGAN [37] | 0.761 |
| AesCLIP [36] | 0.790 |
| **AMHP** | **0.831** |

leverage deep networks to extract visual features (NIMA (Resnet-50) [39], PA_IAA (Densenet-121 [21], MUSIQ [13], TANET [6], Yang *et al.* (Swin-T) [43] and TAVAR [20]). This demonstrates that text templates constructed based on attribute labels can also enable the learned attribute-driven multimodal features to characterize attribute semantic information for image aesthetic quality evaluation more effectively. Compared with the multimodal pre-trained AesCLIP model [36] also based on aesthetic attribute comments, the performance of the proposed AMHP model is improved by 2.4%/2.9%/1.5% in terms of ACC/SRCC/PLCC. This illustrates that our proposed hierarchical prompts learning can more comprehensively capture the semantic information of aesthetic attributes and quality by explicitly simulating users' aesthetic judgments of image quality compared to IAQA models based on multimodal pre-training. Besides, our method also demonstrates that excellent performance can be achieved by constructing text templates of attributes without the availability of user attribute comments on images.

*4.3.3 Performance on the AADB database.* In this database, we also leverage the training images and the corresponding attribute text templates to train the proposed AMHP model and predict the aesthetic quality of the test images. Most of the existing IAQA methods only report their SRCC results for this database. Therefore, we only reported the SRCC results of our AMHP model and state-of-the-art IAQA models. Table 3 summarizes the comparative performance of these methods, and the best performance is shown in bold. As

**Table 4: Performance evaluation of different components of our model on the AVA database.**

| Components | ACC ↑ | SRCC ↑ | PLCC ↑ | MSE ↓ | EMD1 ↓ | EMD2 ↓ |
|---|---|---|---|---|---|---|
| Img | 79.0 | 0.759 | 0.772 | 0.218 | 0.047 | 0.058 |
| Img + Attr | 80.8 | 0.801 | 0.816 | 0.191 | 0.042 | 0.051 |
| Img + Qual | 82.4 | 0.765 | 0.775 | 0.222 | 0.043 | 0.053 |
| Img + Attr + Qual | **84.5** | **0.804** | **0.818** | **0.186** | **0.036** | **0.044** |

listed in Table 3, our model delivers the highest SRCC result, which surpasses the second-best model (AesCLIP) by 4.1%. This also declares that our method is significantly superior to other aesthetic attribute-based IAQA methods. All in all, the above-mentioned performance comparison indicates that the proposed hierarchical prompts learning strategy is very effective in IAQA tasks.

## 4.4 Ablation Study

*4.4.1 Component Ablation.* To verify the effectiveness of the proposed hierarchical prompts, we further explore the contributions of various components of the multimodal feature extraction module to our AMHP model. We train different variants of our model by combining different components and summarized the tested results of these models on the AVA [30] database in Table 4. For each evaluation criterion, the best result is shown in bold font. Specifically, "Img" represents training the proposed model using only images. "Img+Attr" means training the proposed model with both images and attribute comments. "Img+Qual" indicates training the proposed model by utilizing images and text templates of quality levels. "Img+Attr+Qual" represents the complete version of the proposed model.

As listed in Table 4, "Img+Attr" is significantly superior to "Img", which indicates that our model can capture more semantic information about image aesthetic attributes through attribute comments for evaluating image aesthetic quality. Moreover, we find that "Img+Qual" also achieves better results than "Img" on five evaluation criteria except MSE. This indicates that introducing text templates of quality levels has played a positive role in improving the accuracy of our model in measuring aesthetic quality. Furthermore, "Img+Attr" outperforms "Img+Qual" in five evaluation metrics except ACC, indicating that "Img+Attr" performs particularly well in aesthetic score regression and aesthetic distribution prediction, while "Img+Qual" performs better in aesthetic binary classification. Finally, "Img+Attr+Qual" achieves the best results on all evaluation criteria, indicating that each component of our AMHP model contributes to accurately evaluating the aesthetic quality of images. This confirms that the proposed hierarchical prompts learning through attribute comments and quality-level text templates can explicitly simulate users' aesthetic perception process of images, enabling the proposed IAQA model to predict more consistent aesthetic quality with users.

*4.4.2 Attribute Ablation.* In the text template of attributes, the number of aesthetic attributes included also has a significant impact on the proposed model. To verify this viewpoint, we examine the performance of AMHP by using text templates composed of different numbers of aesthetic attributes in the PARA database [43]. The tested SRCC and PLCC results on this database are listed in

**Table 5: Performance evaluation of using different numbers of aesthetic attributes to construct the attribute text template on the PARA database. DOF represents "depth of field" and OB denotes "object emphasis".**

| Attributes | | | | | | SRCC↑ | PLCC↑ |
|---|---|---|---|---|---|---|---|
| Content | Composition | Color | Light | DOF | OB | | |
| ✓ | | | | | | 0.940 | 0.952 |
| ✓ | ✓ | | | | | 0.942 | 0.956 |
| ✓ | ✓ | ✓ | | | | 0.948 | 0.959 |
| ✓ | ✓ | ✓ | ✓ | | | 0.952 | 0.961 |
| ✓ | ✓ | ✓ | ✓ | ✓ | | 0.955 | 0.963 |
| ✓ | ✓ | ✓ | ✓ | ✓ | ✓ | **0.955** | **0.966** |

Table 5 and the best results are shown in boldface. This table shows that our model can achieve relatively satisfactory performance (compared to other methods in Table 2) even when text templates contain only one attribute. In addition, as the number of attributes included in text templates increases, the overall performance of our model also continues to improve. This indicates that the diversity of attributes in the attribute text templates promotes the proposed model to learn comprehensive semantic information about aesthetic attributes, which in turn enables more accurate judgments of image aesthetic quality. Similar conclusions can also be drawn from ablation experiments on attributes in the AADB database [17]. Therefore, to make the constructed attribute text templates concise and effective, we adopt three different aspects of attributes (color harmony, light, and content) to construct text templates, which can also achieve satisfactory performance in Section 4.3.3.

## 4.5 Visual Analysis

To intuitively demonstrate the performance of the proposed AMHP in learning users' aesthetic perception of images, we conduct a visual analysis of our model for assessing image aesthetic quality. Fig. 3 shows randomly selected two test images from the AVA, PARA, and AADB databases respectively. For the AVA database, we present user comments on the attributes of the images. For the PARA and AADB databases, we show the corresponding attribute text templates constructed using the image attribute labels. To verify the effectiveness of hierarchical prompts learning, we compare our AMHP model with the CLIP model that only adopts quality levels for prompt learning and show the normalized ground-truth (GT) quality scores of images and the predicted quality scores by the above two models.

As shown in Fig. 3, we can see that the CLIP model using only quality level prompts has difficulty in accurately predicting the aesthetic quality scores of images. In contrast, our proposed hierarchical prompts learning can more precisely evaluate the aesthetic quality scores of images. The underlying reason is that the proposed attribute-driven multimodal features are efficient in representing the semantic information of image aesthetic attributes. Furthermore, by leveraging the proposed hierarchical prompts learning to explicitly model users' aesthetic judgment process of images, a superior performance IAQA model is obtained through the joint prompts of semantic information related to aesthetic attributes and quality levels.

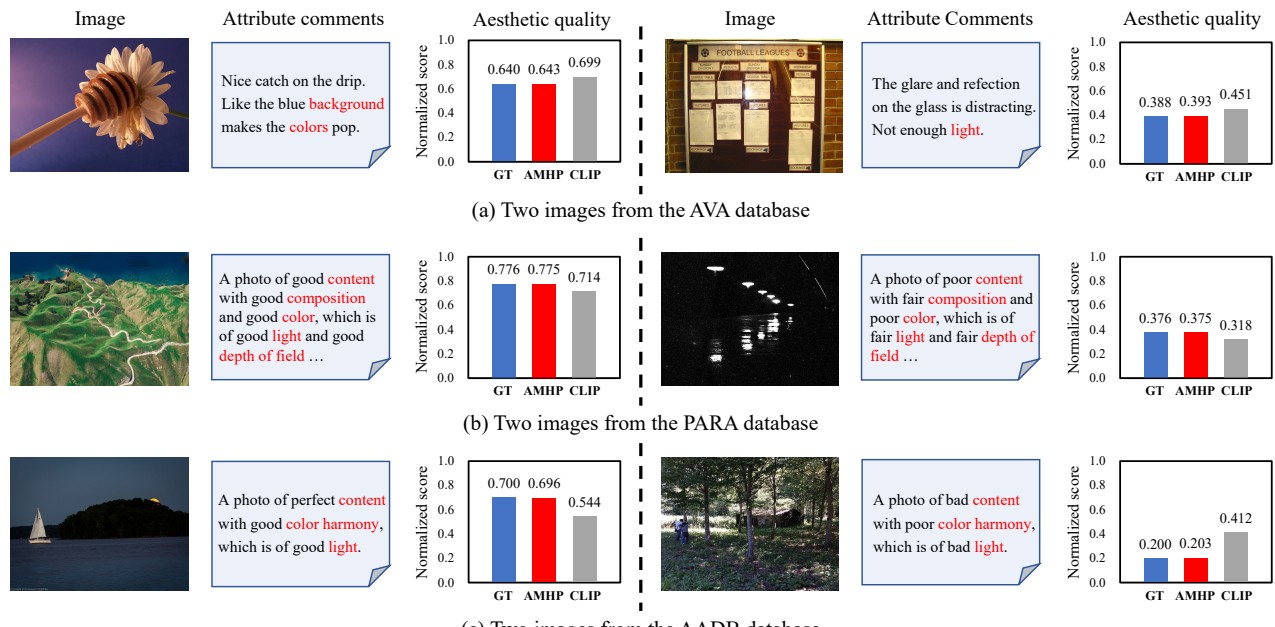

Figure 3: Two test images were randomly selected from the (a) AVA, (b) PARA, and (c) AADB databases, respectively. On the right side of each image, we also present the attribute comments of users (AVA) or the constructed attribute text templates (PARA and AADB). In addition, the normalized ground-truth (GT) quality scores of the images, as well as the predicted quality scores through our AMHP method and CLIP model, are also shown.

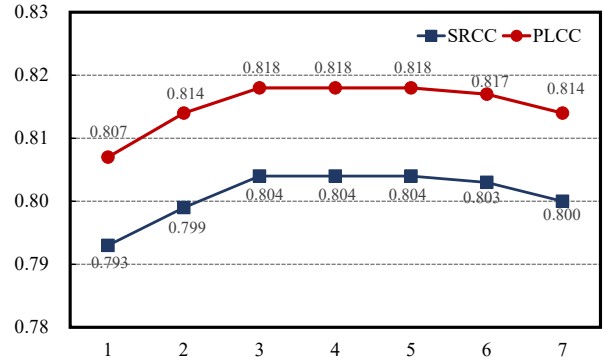

Figure 4: Performance changes of SRCC and PLCC for test images during the training of our model on the AVA database.

## 4.6 Discussion

In this section, we discuss the convergence ability of the proposed hierarchical prompts learning during the training process. Fig. 4 shows the trends in the changes of SRCC and PLCC for test images during the training of our AMHP model on the AVA [30] database. As shown in the figure, the proposed model can achieve promising performance after just one epoch. As the number of epochs increases, the results of SRCC and PLCC also improve accordingly. After the third epoch, the results of both SRCC and PLCC remain consistent within two epochs and then begin to decrease slightly, which indicates that our model can converge quickly. We have also conducted relevant experiments on the PARA and AADB databases,

and similar conclusions can be drawn from the experimental results. Therefore, the proposed multimodal hierarchical prompts learning not only allows our model to achieve efficient performance but also saves computational resources during the training process.

## 5 CONCLUSION

In this paper, we have presented an attribute-driven multimodal hierarchical prompts (AMHP) method for image aesthetic quality assessment. To effectively reveal the implicit information in users' aesthetic judgments of images, we propose a multimodal hierarchical prompts learning approach, which gradually captures the semantic features of aesthetic attributes and quality to perform prompt learning for image aesthetic quality. Firstly, the proposed attribute-driven multimodal features have been proven to effectively represent the semantic information of users' perception of image aesthetic attributes. Then, further introducing text templates for different quality levels can enable the proposed model to more precisely evaluate the aesthetic quality of images. In summary, the proposed method can leverage the hierarchical prompts of semantic information related to aesthetic attributes and different quality levels, explicitly simulating users' aesthetic judgments of images, and resulting in a more efficient IAQA model. Experimental results on multiple IAQA databases indicate that our AMHP model achieves better performance than state-of-the-art IAQA methods in evaluating the aesthetic quality of images, providing insights into modeling the perception process of users' aesthetic judgments through multimodal prompt learning.

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
