# OpenReview forum: "Attribute-Driven Multimodal Hierarchical Prompts for Image Aesthetic Quality Assessment"
_acmmm.org/ACMMM/2024/Conference — MM2024 Poster_

### Official Review · Reviewer_LBgq · 2024-05-19

**Rating:** 5
**Confidence:** 4

**Summary:**

This paper proposes an image aesthetic quality assessment method based on attribute-driven multimodal hierarchical prompts. The proposed method leverages attribute comments and quality level text templates to hierarchically learn the aesthetic attributes and quality of images.

**Strengths:**

(1)The author propose an attribute-driven multimodal learning strategy. By embedding aesthetic attribute comments with images, they can utilize more comprehensive semantic information of aesthetic attributes for modeling the aesthetic quality of images. It is novelty.
(2)The author propose a multimodal hierarchical prompts learning approach. By hierarchically utilizing text descriptions of aesthetic attributes and different quality levels for multimodal prompted learning, which is also novelty.

**Limitations:**

(1)In Figure1 “Users first evaluate the aesthetic attributes of images through textual comments. Then, users can explicitly judge the aesthetic quality of images based on these attribute comments. ” does not cover the images  which do not have  " attribute comments."
(2)In line178-180, "the proposed method can effectively simulate users’ aesthetic judgments of images in actual situations.".Besides,  in Figure 2 ” simulate users’ aesthetic quality judgment of images.” Is there any  basis for this statement? When we judge an image , we usually judge  an image  without the attribute  comments.
(3)For AVA datasets, in the testing phase, is it compulsory to rely on “Attribute Comments ” for evaluate the image aesthetic quality?  Besides，in Table4, Performance evaluation of different components of our model on the AVA database,  For Img , Img + Attr，Img + Qual，Img + Attr + Qual，in the testing phase ，what are the inputs？
(4)For  PARA and  AADB, in the tesing phase,  except for image as input ,  is it compulsory to rely on “text templates of the corresponding attributes ” for evaluate the image aesthetic quality?
(5) In line 606-607， “Therefore, we can leverage attribute comments to train and test our AMHP model.” The test dataset for AVA is not strictly AVA image, but AVA-Comments. You use comments as supplement information  while other models  only use image. The model you proposed should be called   image-comments  aesthetic quality evaluation ,not image aesthetic quality evaluation in traditional sense. So the results in Table4 is the results based on the comments,which have image aesthetic semantic information,which is not appropriate comparing with the methods without comments.
(6)In line 403. ” We assume that 𝑘 represents the number of total quality levels. “ How to choose the most appropriate parameter 𝑘    for different datasets?

**Suitability:**

3

---

### Official Review · Reviewer_TkaF · 2024-05-20

**Rating:** 3
**Confidence:** 4

**Summary:**

This paper describes a method for Image Aesthetic Quality Assessment (IAQA) that aims to mimic users' visual perception to evaluate the aesthetic quality of images. Unlike existing methods that mainly use multi-modal pre-training or direct hints for model learning, this method adopts attribute annotations and quality-level text templates to hierarchically understand the aesthetic attributes and quality of images, providing a novel approach to IAQA. This enables a more accurate assessment of aesthetic quality compared to existing technologies.

**Strengths:**

1.	This paper proposed a new IAQA method by combining attribute-driven multi-modal hierarchical prompts, which has good innovation points.
2.	The effectiveness of the method is better proved by combining the review text with image aesthetic evaluation and rich experiments.

**Limitations:**

1.	The AVA-Comments database does not seem to have attribute tags. The model uses attribute keywords for classification. What are these keywords and how are they derived? Is it a self-constructed vocabulary list or some other method？
2.	Some comments describe multiple aesthetic attributes. How are the models classified?
3.	Some comments in the AVA-Comments database have both good and bad comments. How to set the quality level text template in the model? Is it reasonable to use cosine similarity at this time?
4.	AADB and PARA databases do not contain comments, and the model replaces comments by automatically constructing corresponding attribute text templates. Whether automatic construction is effective should be demonstrated in ablation experiments.

**Suitability:**

3

---

### Official Review · Reviewer_oXhd · 2024-05-23

**Rating:** 3
**Confidence:** 3

**Summary:**

This paper proposes multimodal prompts based on the pre-trained visual language model, CLIP, to address the issues of automatic aesthetic quality assessment. Specifically, the authors propose two types of prompts. The first-level prompts integrate aesthetic attribute comments and image feature, while the second-level prompts focus on determining the aesthetic quality level of the images. Experimental results have demonstrated the effectiveness of the proposed modules. However, there are several issues that need to be addressed. Detailed comments are as follows.

**Strengths:**

1.	The introduction of hierarchical prompts is well-aligned with the IAQA task, simple but effective.
2.	The method presents comparative experimental results, and ablation studies also validate the effectiveness of the proposed components.
3.	The paper is well-written and logically structured.

**Limitations:**

1.	The authors emphasize the importance of hierarchical prompts, suggesting that the hierarchical nature of attribute-driven multimodal prompts and the text prompts representing quality can improve model performance. However, the authors have not fully demonstrated this point. Maybe, they should concatenate the two types of prompts into the "flat prompts" for comparison with the "hierarchical prompts", which are proven effective in many tasks.
2.	In Equation 5, the motivation for the authors to add the multimodal features obtained through the FC layer to the image features as the final multimodal features is not very clear. In fact,
𝑅_xt is already a multimodal feature. If further processing is needed, shouldn't residual connection techniques be used instead? Moreover, it is also worth considering whether adding text features here still benefits the results.

**Suitability:**

3

---

### Meta-Review · Area_Chair_r8Xq · 2024-06-28

**Recommendation:** Accept (Poster)
**Confidence:** 4

**Metareview:**

The reviewers are unanimous in the opinion that the paper is very relevant to the scope of ACM Multimedia. They are also positive about certain novel aspects of the proposed approach, such as the introduction of hierarchical multimodal prompts. However, they have identified certain limitations, such as regarding the evaluation protocol.

Taking into account reviewer comments and the rebuttal process, I am leaning towards acceptance of the paper.